# Systematic review of food insecurity and violence against women and girls: Mixed methods findings from low- and middle-income settings

Abigail M. Hatcher [1,2]*, Sabrina Page[3,4], Lele Aletta van Eck[2], Isabelle Pearson[4], Rebecca Fielding-Miller[5], Celine Mazars[6], Heidi Stöckl[3,4]

1 Galling's School of Global Public Health, University of North Carolina, Chapel Hill, Chapel Hill, North Carolina, United States of America, 2 School of Public Health, Faculty of Health Sciences, University of the Witwatersrand, Johannesburg, South Africa, 3 Institute for Medical Information Processing, Biometry, and Epidemiology, Ludwig-Maximilians-University, Munich, Germany, 4 Department of Global Health and Development, London School of Hygiene and Tropical Medicine, London, United Kingdom, 5 Herbert Wertheim School of Public Health, University of California, San Diego, San Diego, California, United States of America, 6 Independent Researcher, Paris, France

* abbeymae@email.unc.edu

**Data Availability Statement:** Quantitative data is included within the manuscript (Table 1 and Fig 2).

## Abstract

Violence against women and girls (VAWG) is a global human rights and public health concern. Food insecurity is a sign of severe poverty, and likely to heighten women's vulnerability to VAWG and men's perpetration of it. However, the extent of the association and the multiple pathways between food insecurity and VAWG are not well understood. We systematically assessed peer reviewed quantitative and qualitative literature to explore this in low- and middle-income countries. Fixed effects meta-analysis was used to synthesize quantitative evidence. Qualitative data was analyzed using thematic analysis. From a search of 732 titles, we identified 23 quantitative and 19 qualitative or mixed-methods peer-reviewed manuscripts. In a meta-analysis of 21 cross-sectional studies with 20,378 participants, food insecurity was associated with doubled odds of reported VAWG (odds ratio [OR] = 2.38, 95% confidence interval [CI] = 1.82–3.10). This finding was consistent for both women's experience or male perpetration of VAWG. Qualitative and mixed-methods papers offered insight that underlying conditions of inequitable gender norms, economic deprivation, and social isolation frame both food insecurity and VAWG. Food insecurity may trigger survival behaviors due to household stress and lack of meeting expected gender roles, which leads to VAWG. VAWG exposure may lead to food insecurity if women are more impoverished after leaving a violent household. Potential protective factors include financial stability, the involvement of men in VAWG programming, transformation of gender norms, and supporting women to develop new networks and social ties. Strong evidence exists for a relationship between food security and VAWG. Future funding should target causal directions and preventive options through longitudinal and interventional research. Strategies to ensure households have access to sufficient food and safe relationships are urgently needed to prevent VAWG.

Qualitative data is available online in original papers.

**Funding:** The authors received no specific funding for this work.

**Competing interests:** The authors have declared that no competing interests exist.

## Introduction

The World Health Organization estimates that globally, one in four women experience violence against women (VAWG) in their lifetime [1]. In sub-Saharan Africa, 44% of women experience VAWG from an intimate partner and one in six experience non-partner violence [2]. VAWG forms a persistent health crisis, leading to a high burden of injury, mental illness, physical decline, and mortality [3, 4]. VAWG is also a human rights travesty that circumvents governmental and bilateral commitments and undermines Sustainable Development Goals [5].

Emerging evidence suggests food insecurity may be one factor related to VAWG. Food insecurity is defined as having uncertain or limited availability of nutritionally adequate food or the inability to acquire safe, acceptable foods [6]. Beyond sheer hunger from insufficient food intake, food insecurity also includes poor dietary quality and worry or anxiety over securing food supplies [7]. It can incorporate social and psychological elements of shame or status in a community. In 2019, an estimated 1.3 billion people lacked regular access to food [8], and this number is likely to have increased since the SARS Cov-2 pandemic (Covid-19) [9]. Food security is also worsening due to global climate change, with droughts leading to insufficient agricultural output [10].

Being food insecure seems to be related to women's exposure to violence, though the pathways for this are poorly understood. Meinzen-Dick and colleagues posit that safety from VAWG improves women's household bargaining power, thereby improving food security [11]. Hatcher *et al.* theorized that food insecurity leads to VAWG perpetration by prompting conflict, relationship control, and causing increased alcohol intake [12]. Cash transfer trials have found that eliminating food insecurity may reduce household conflict and improve decision-making [13]. It is plausible that a bidirectional relationship between these two conditions may exist.

The evidence on food insecurity and VAWG has yet to be brought together systematically. The gaps around the links between food insecurity and VAWG as well as potential solutions are crucial to fill if we are to inform future programs and sustainable development goals.

## Methods

We conducted a mixed-methods systematic review. Using the quantitative evidence, we performed a meta-analysis to estimate the effect of the association between food insecurity and VAWG in populations living in low- and middle-income countries (LMIC). We analyzed the qualitative evidence using thematic analysis and synthesized the qualitative and mixed methods evidence to explore the drivers of food insecurity and VAWG, with an emphasis on possible pathways linking these conditions. The methodology for this systematic review was developed in line with the Preferred Reporting Items for Systematic Reviews and Meta-Analyses (PRISMA) [9].

### Search strategy

We searched five databases (PubMed, Web of Science, CINAHL, Global Health, and PsycInfo) from January 2000 through to July 2021. A full search strategy included key words and Medical Subject Headings (MESH) terms around the four constructs (S1 Text): food insecurity, VAWG, NOT plants or animals and LMIC settings.

### Eligibility criteria

Inclusion and exclusion criteria were constructed and applied to all results that came up relative to the search terms. The search terms can be found in S1 Text. Studies were limited to original research published in peer-reviewed journals in the English language in low-and

middle-income countries. The review included quantitative, qualitative, and mixed methods interventional or observational study designs.

Studies must have also reported the association between food insecurity and VAW among adult populations, defined as aged 15 and above, with the study setting in a LMIC based on current World Bank rating. Food insecurity is defined as the situation in which an individual or household has difficulties accessing sufficient, safe, culturally appropriate and nutritious food to meet dietary requirements and preferences for a healthy life due to lack of money or other resources [4]. Qualitative studies were included if they used focus groups and/or interviews to assess experiences with food insecurity and access [13]. Studies were excluded if they did not include a food insecurity measure as an exposure for VAWG or if they did not assess food insecurity and VAWG together.

*Study selection* followed a three-step process: title and abstract review, full text review, and quality appraisal. First, multiple authors (AMH, SP, LvE, IP) reviewed all identified study titles and abstracts using a double-blind process. Duplicates and studies that did not meet the inclusion criteria were removed. The same authors independently assessed the full papers of those abstracts that met the eligibility criteria.

Finally, the authors conducted a *quality appraisal* on all full texts using the Newcastle–Ottawa Scale (NOS) for observational studies [14]. NOS is a tool to assess the quality of non-randomized studies to be used in a systematic review. Each study is judged with a 'star system' on three points: the selection of the study groups, the comparability of the groups, and the ascertainment of the exposure or outcome.

At each stage of the process, reviewers assessed discrepancies and garnered input from the senior authors (CM and HS) until a consensus was reached. Finally, the three authors manually searched the reference lists of the included articles for further key studies that could potentially be included in the analysis.

**Data abstraction.**    The following data was extracted and summarized in tables: citation; year of publication; country; study design and sampling; characteristics of the study population; outcomes (VAWG type); measures used (for food insecurity and VAWG). When available, adjusted odds ratios (aOR) and similar estimates (e.g. relative risk, hazard ratio) with confidence intervals were extracted. Additionally, count data was extracted in a two-by-two table to be interpreted as an odds ratio. When count data was not presented in the published manuscript, authors were contacted by email to invite them to share a simple two-by-two table.

**Data analysis.**    Quantitative outcomes were extracted into an Excel table. Pooled unadjusted odds ratio (OR) estimates were calculated using random effects meta-analysis *metan* command in STATA 12.0 [15]. Sub-analyses were conducted to assess meta-analytic findings by sex (experience among women or perpetration by men), VAWG type (physical vs. sexual vs. emotional), and geography of study setting.

Qualitative data were thematically coded by summarizing themes from each included qualitative paper [16]. These themes were grouped inductively by three researchers (AMH, CM, SP). We highlighted any geographical or methodological gaps in the current literature. In a section on mixed-methods findings, we explored what the current literature says around mechanisms on *why* food insecurity and IPV might be related? Here we drew upon available quantitative data that highlights mediators or directionality between the two conditions.

## Results

### Study characteristics

Our original search yielded 742 titles. Including seven papers found by contacting authors in the field, a total of 38 papers were included in the systematic review (Fig 1).

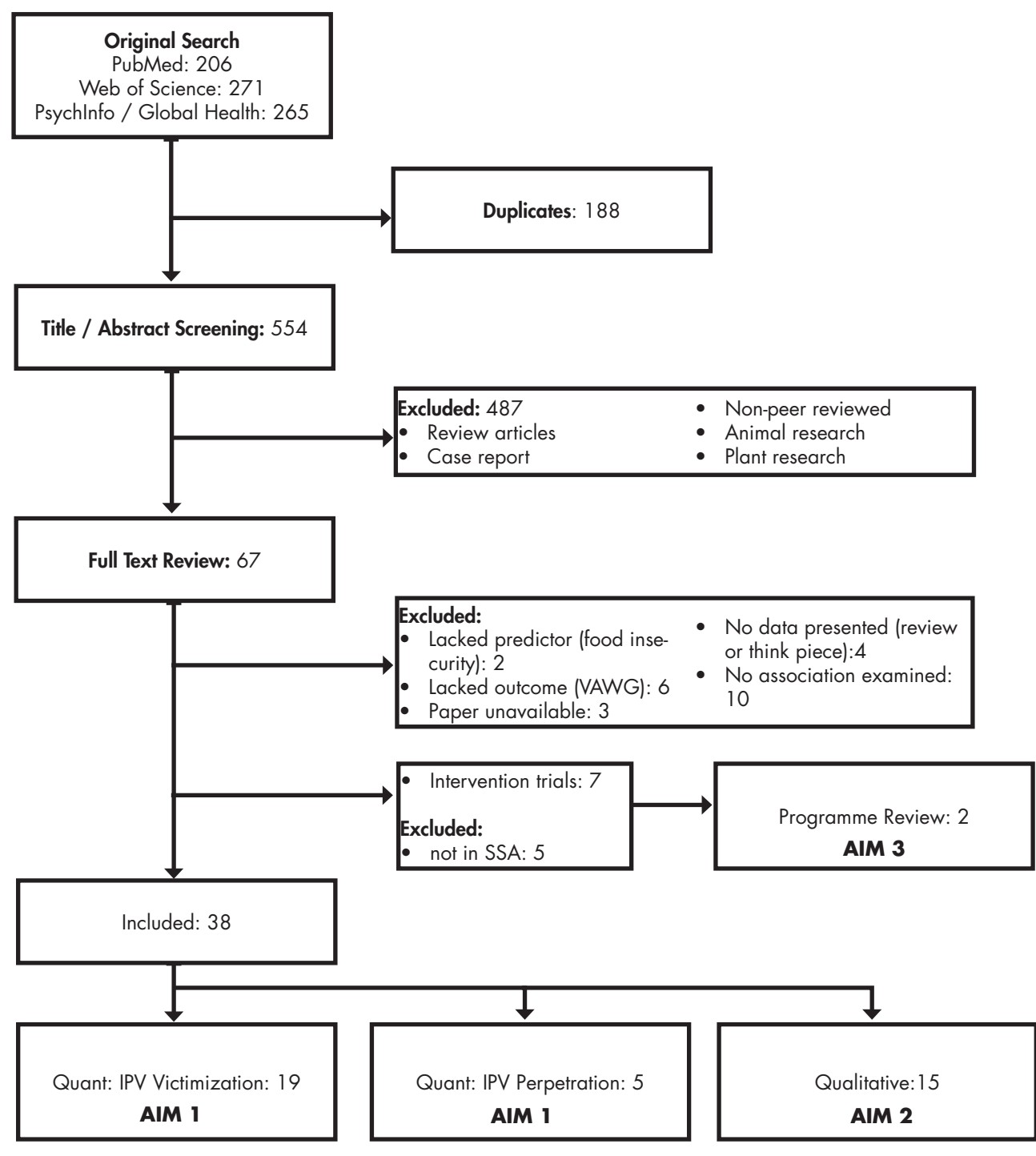

**Fig 1. Study selection flow diagram.**

Our search identified 23 studies that examined the quantitative association between food insecurity and VAWG (Table 1). The majority of studies (*n* = 20) examined VAWG experience among women, while five studies looked at men's VAWG perpetration. Studies were published from 2012–2020 and were conducted in Africa (Cote d'Ivoire, Ethiopia, Kenya, Liberia, South Africa, Eswatini, Uganda), Asia (Afghanistan, Bangladesh, Khazakstan, Nepal), and

**Table 1. Quantitative studies included in systematic review.**

| Author | Year | Country | Group | Study type | Sample size | Food insecurity | Measure | VAWG | Measure | Time period | Type of VAW | Analysis | Adjusted outcome | Control variables |
|---|---|---|---|---|---|---|---|---|---|---|---|---|---|---|
| Andarge, | 2018 | Ethiopia | Women | cross-sectional | 737 | Food insecurity | food insecurity experience scale | IPV experience | WHO Instrument | 12 months | Phys, Sex, Emot | Logistic Regression | aOR = 6.59 (4.54–9.57) | age of woman, partner age difference, decision-making |
| Barnett | 2019 | South Africa | Pregnant women | cross-sectional | 992 | Food insecurity | USDA Household Food Security Scale | IPV experience | WHO Instrument | 12 months | Phys, Sex, Emot | Logistic Regression | aOR (emotional) = 1.60 (1.04 to 2.46) | income, education, community, childhood trauma, stressful life events, depression, distress |
| de Moraes | 2016 | Brazil | Women | cross-sectional | 845 | Food insecurity | Brazilian version of Household Food insecurity Access Scale (HFIAS) | IPV experience | Revised conflict tactic scale | 12 months | Phys, Psych | Path Analysis | adjusted outcome data not presented | |
| Diamond-Smith, | 2019 | Nepal | Women | cross-sectional | 3373 | Food insecurity | Household Food insecurity Access Scale (HFIAS) | IPV experience | Demographic Health Survey items | 12 months | Phys, Sex, Emot | Logistic Regression | aOR = 2.48 (1.52–4.04) | age, education, ethnicity, geographic region, household head, living with partner, occupation, alcohol, household wealth |
| Field, | 2018 | South Africa | Pregnant women | cross-sectional | 376 | Food insecurity | Household Food Security survey module | IPV experience | Revised conflict tactic scale | 6 Months | Phys, Sex, Emot | Logistic Regression | aOR = 2.43 (1.50–4.57) | age, relationship type, social support, past abuse, mental health |
| Fielding-Miller, | 2015 | Swaziland | Pregnant women | cross-sectional | 405 | Food insecurity | 7 items | IPV experience | WHO Instrument | 12 months | Phys / Sex | Logistic Regression | adjusted outcome data not presented | |
| Fielding-Miller, | 2020 | Swaziland | Young, university women | cross-sectional | 372 | Food insecurity | 7 items | Sexual assault | Sexual experience Survey Short Form | 12 months | Sexual | Logistic Regression | aOR = 2.43 (1.50–4.57) | ever had a child, bursary as support, childhood abuse exposure |
| Fong | 2016 | Cote d'Ivoire | Women | cross sectional | 68 | Food insecurity | HFIAS | IPV experience | unknown | 12 months | | Logistic Regression | aOR = 8.36 (2.29–30.57) | children, marital status |
| Gibbs, | 2018 | Afghanistan | Women | cross sectional | 935 | Food insecurity | Household Hunger Scale (HHS) | IPV experience | WHO Instrument | 12 months | Phys, Emot | Multinomial Regression | aRRR = 1.13 (1.03–1.25) | age, cluster, education, ethnicity, gender attitudes, childhood trauma, earnings, polygyny, family violence, depression, disability |

*(Continued)*

**Table 1.** (Continued)

| Author | Year | Country | Group | Study type | Sample size | Food insecurity | Measure | VAWG | Measure | Time period | Type of VAW | Analysis | Adjusted outcome | Control variables |
|---|---|---|---|---|---|---|---|---|---|---|---|---|---|---|
| Gibbs, | 2018 | South Africa | Young women and men in informal settings | cross sectional | 1357 | Food insecurity | HHS | IPV perpetration/ experience | WHO Instrument | 12 months | Phys / Sex | Logistic Regression | aoR (women) = 1.84 (1.08–3.14) aOR (men) = NS; | age, education, intervention arm, relationship status, quarreling, controlling behaviours, alcohol use, depressive symptoms |
| Gilbert, | 2019 | Khazakstan | Male migrant workers | cross-sectional | 1342 | Food insecurity | Single item | IPV perpetration | Revised conflict tactic scale | 6 Months | Phys / Sex | Logistic Regression | aOR (external migrant) = 4.37 (1.72–11.07) | age, marital status, children, religion, ever in jail, childhood sexual abuse, alcohol, social support, depression |
| Hatcher, | 2019 | South Africa | Men in informal settings | cross sectional | 2006 | Food insecurity | HHS | IPV perpetration | WHO Instrument | 12 months | Phys / Sex | Logistic Regression | aOR = 2.18 (1.75, 2.54) | migrancy, eduction, age, unemployment |
| Jewkes, | 2019 | Afghanistan | Women | cross-sectional | 1462 | Food insecurity | 3 items | IPV experience | WHO Instrument | 12 months | Physical | Multinomial Regression | adjusted outcome data not presented | |
| Logie, | 2019 | Uganda | Displaced young women | cross-sectional | 233 | Food insecurity | Single item | IPV experience | 3 Items | 12 months | Phys, Sex, Emot | Multinomial Regression | aOR (>1 type VAWG) = 7.15 (1.32, 38.89) | age, relationship status, mobile phone ownership, depression, childhood violence, transactional sex, sexual relationship power, community safety |
| Naved | 2018 | Bangladesh | Women | cross-sectional | 800 | Food insecurity | 3 items | IPV experience | WHO Instrument | 12 months | Phys / Sex | Logistic Regression | aOR (phys) = 3.78 (1.29–11.19) | Age, education, children, income, savings, household contribution, acceptance of IPV, control or fights by husband |
| Orindi | 2020 | Kenya | Young women | cross-sectional | | Hunger | Single item | IPV experience | WHO Instrument | 6 months | Phys, Sex, Emot | Logistic Regression | aOR (phys) = 1.38 (1.01–1.89) | age, intervention arm, site, marital status, in school, religion, sexually active |

*(Continued)*

**Table 1.** (Continued)

| Author | Year | Country | Group | Study type | Sample size | Food insecurity | Measure | VAWG | Measure | Time period | Type of VAW | Analysis | Adjusted outcome | Control variables |
|---|---|---|---|---|---|---|---|---|---|---|---|---|---|---|
| Regassa | 2012 | Ethiopia | Women | cross-sectional | 1094 | Food insecurity | HFIAS | IPV experience | WHO Instrument | 12 months | Phys, Sex, Emot | Logistic Regression | aOR = 2.18 (1.75, 2.54) | age, age difference, marital status, education, literacy, religion, household size, alcohol use |
| Rahman, | 2013 | Bangladesh | Women | cross-sectional | 3861 | Chronic undernutrition | BMI<18.5Kg/m2 | IPV experience | Revised conflict tactic scale | 12 months | Phys / Sex | Logistic Regression | aOR = 1.22 (1.04–1.43) | age, education, decision-making autonomy, occupation, religion, residence, number of household members, ever use of contraception and respondent's height |
| Ribeiro-Silva, | 2016 | Brazil | Poor families | cross-sectional | 1019 | Food insecurity | Brazilian HFIAS | IPV perpetration | Revised conflict tactic scale | 12 months | Physical | Logistic Regression | aPR (minor violence) = 1.23 [1.12, 1.35] | economic status, income, agglomeration and education level |
| Schneider, | 2018 | South Africa | Pregnant women | cross-sectional | 425 | Food insecurity | HFIAS | IPV experience | 2 item quesionnairre | 3 Months | Phys / Sex | Bivariate | adjusted outcome data not presented | |
| Shai, | 2019 | Nepal | Migrant workers | cross-sectional | 357 | Hunger | Single item | IPV perpetration/experience | WHO Instrument | 12 months | Phys, Sex, Emot | Logistic Regression | aOR (older women) = 1.77 (1.11–2.81) aOR (men) = NS | age, depression, mother-in-law relationship |
| Swahn, | 2015 | Uganda | Young women | cross-sectional | 313 | Hunger | Single item | Sexual assault | Youth risk behaviour survey, | 12 months | Sexual | Logistic Regression | aOR = 3.73 [1.92, 7.22] | age, alcohol. Loneliness |
| Willie, | 2018 | Liberia | Pregnant women | cross-sectional | 195 | Food insecurity | 1 item from Harvard Trauma Questionnaire | IPV experience | Revised conflict tactic scale | Lifetime | Phys / Sex | Logistic Regression | aOR = 2.55 [1.32, 4.94] | age, education, employment, relationship status, children |

South America (Brazil). All included studies were cross-sectional reports. Of the 18 qualitative and mixed-methods papers included in the synthesis, 9 took place in sub-Saharan Africa, 4 in Asia, and 2 in Latin America (Table 1).

## Measurement

Overall, measurement of food insecurity and VAWG in included studies was of high quality. Most studies used self-reported measures to assess food insecurity (with one study objectively assessing malnutrition by measuring body mass index [17]). All studies used self-reported measures of VAWG. While self-reports represent the state of art for food insecurity and VAWG research, there are clear limitations in reporting bias for this type of measure, such as underreporting and social desirability bias.

Multiple studies used comprehensive measures of food security. The Household Food Insecurity Access Scale (HFIAS) and the Brazilian Food Insecurity Scale (BFIS) measure constructs of quality and quantity of food supply as well as anxiety around food supply. Four studies used HFIAS [18–21] and two used BFIS [22, 23]. Another strong measure is the USDA Household Food Security Scale [24]. Three papers used 3-items of HFIAS related to household hunger (called the Household Hunger Scale [HHS]) [12, 25, 26]. Several studies used a single item to assess food insecurity [27–31], which may increase the risk of misclassification.

Measurement of VAWG varied but a majority used validated scales that assessed between 9 to 14 behaviourally-specific actions to classify whether violence occurred or not. Tools included the World Health Organization multi-country study instrument [12, 20, 24–26, 29, 30, 32–35], the Revised Conflict Tactics Scale [17, 22, 23, 27, 36, 37], the Sexual Experiences Survey [33], and the Demographic Health Survey Domestic Violence module [18]. Three studies used non-validated 1 to 3 single item questions asking participants whether they experienced any physical or sexual intimate partner? violence [21, 28, 31].

## Ethical considerations of studies

Many studies included an explanation of the ethical procedures undertaken if a participant disclosed experience of or perpetration of VAWG. While this was not a focus of the review, it is important to note that this ethical section is required by most experimental trial reporting and represents best-practice in the violence research field.

## Association between food insecurity and VAWG

Eighteen studies examined the association between food insecurity and VAWG experience among women. The majority of these studies were among adult women in the general population. Five studies were conducted among women aged 18+ [26, 28, 29, 31, 33] and four among pregnant adult women [21, 36, 37]. Shai et al. examined migrant workers [30], while Naved et al. conducted research among factory workers [35]. The young women Logie et al. interviewed were displaced by migration [28]. Every study reporting adjusted associations of food insecurity and VAWG experience among women found statistically significant results. The odds ratios ranged from 1.22 to 8.36, suggesting that the two conditions are significantly and strongly associated in cross-sectional data.

Five studies included measures of men's perpetration of VAWG [12, 23, 26, 27, 30]. Four studies were among adult men in the general population, though Gibbs et al. conducted research among adult men aged 18–30. Gibbs et al. and Hatcher et al. conducted research in informal, urban settings [26] while Gilbert et al. and Shai et al. focused on migrant communities [27, 30]. Three studies reported significant associations between food insecurity and VAWG perpetration by men [12, 23, 27]. The adjusted odds ratios for studies reporting a significant association ranged from 2.18–4.37. The two studies in South Africa and Nepal that did

not report a significant association presented no data on the adjusted relationship between food insecurity and VAWG perpetration [26, 30].

## Meta-analytic findings

The meta-analysis included 21 studies with relevant bivariate data (Table 2). Overall, the relationship between food security and VAWG was significant (Fig 2). Being food insecure doubled the odds that a participant reported VAWG exposure (odds ratio [OR] = 2.38, 95% confidence interval [CI] = 1.82–3.10).

When examined by sex, the meta-analytic findings stay consistent (Fig B1 in S1 Fig). Studies examining women's VAWG reporting found that food insecurity more than doubles the odds of them experiencing VAWG (OR = 1.98, 95%CI = 1.79–2.18). If men are food insecure, they report roughly double the odds of perpetrating VAWG (OR = 1.90, 95%CI = 1.63–2.22).

Results vary somewhat by VAWG type, though this should be tempered by the limited number of studies in each category (Fig B2 in S1 Fig). When studies measured VAWG as "any physical, sexual or emotional form of violence", food insecurity more than doubled the odds of violence (OR = 2.27, 95%CI = 1.89–2.72). When studies measured it as "physical violence only", the association was similar and slightly stronger (OR = 2.55, 95%CI = 2.17–3.01). By contrast, using a measure of "any physical and/or sexual violence" led to 70% increased odds of VAWG (OR = 1.70, 95%CI = 1.55–1.87). The number of studies included in "sexual only" (*n* = 2) or "physical and/or emotional" (*n* = 1) VAWG were too small to be interpreted, though both were above three-fold odds.

The meta-analytic results vary slightly by region (Fig B3 in S1 Fig). African studies show a stronger association between the two conditions (OR = 2.17, 95%CI = 1.97–2.39) compared to Asian studies (OR = 1.86, 95%CI = 1.67–2.09). In terms of timeframe (Fig B4 in S1 Fig), studies conducted before *El Nino* in 2016 show a weaker association between food insecurity and

**Table 2. Qualitative and mixed-methods studies included in systematic review.**

|  | Author | Year | Country | Population group | Data collection method | Sample size |
|---|---|---|---|---|---|---|
| 1 | Ager, | 2018 | Uganda | Women, Men, Boys and girls | Focus Groups | 64 Groups |
| 2 | Bellows, | 2015 | Georgia & South Africa | States | Case study | n/a |
| 3 | Bonatti, | 2019 | Tanzania | Men, Women | Survey/ Workshop | 333 |
| 4 | Buller | 2016 | Ecuador | Men, Women | Qual interviews + Focus Groups | 48 IDIs; 8 FGDs |
| 5 | Buller | 2018 | Global | Peer-review publications | Lit Review | n/a |
| 6 | Cardoso, | 2016 | Côte d'Ivoire, | Men and Women | Focus group | 91 |
| 7 | Davis, | 2018 | Zambia | Men and Women | Survey | 204 |
| 8 | Derose, | 2017 | Dominican Republic | Women with HIV | Qual interviews | 30 |
| 9 | Deuba, | 2016 | Nepal | Pregnant women | Qual interviews | 20 |
| 10 | De Moraes | 2016 | Brazil | Women | Quantitative | 849 |
| 11 | Fielding-Miller | 2019 | Eswatini | Women victims of VAWG | Qual interviews | 20 |
| 12 | Hatcher | 2019 | South Africa | Men | Quantitative | 2604 |
| 13 | Lemke, | 2003 | South Africa | Men and Women | Qual interviews | 166 |
| 14 | Lentz, | 2019 | South Asia | Women | Qual interviews | 134 |
| 15 | Meinzen-Dick | 2019 | Global | Peer-review publications | Lit Review | n/a |
| 16 | Miller, | 2011 | Uganda | Women & Men | Qual interviews | 41 |
| 17 | Roy | 2019 | Bangladesh | Women | Qual interviews | not stated |
| 18 | Sethuraman, | 2006 | South India | Women & children | Focus group | 820 |
| 19 | Zakaras | 2017 | Kenya | Women & Men | Qual interviews | 54 |

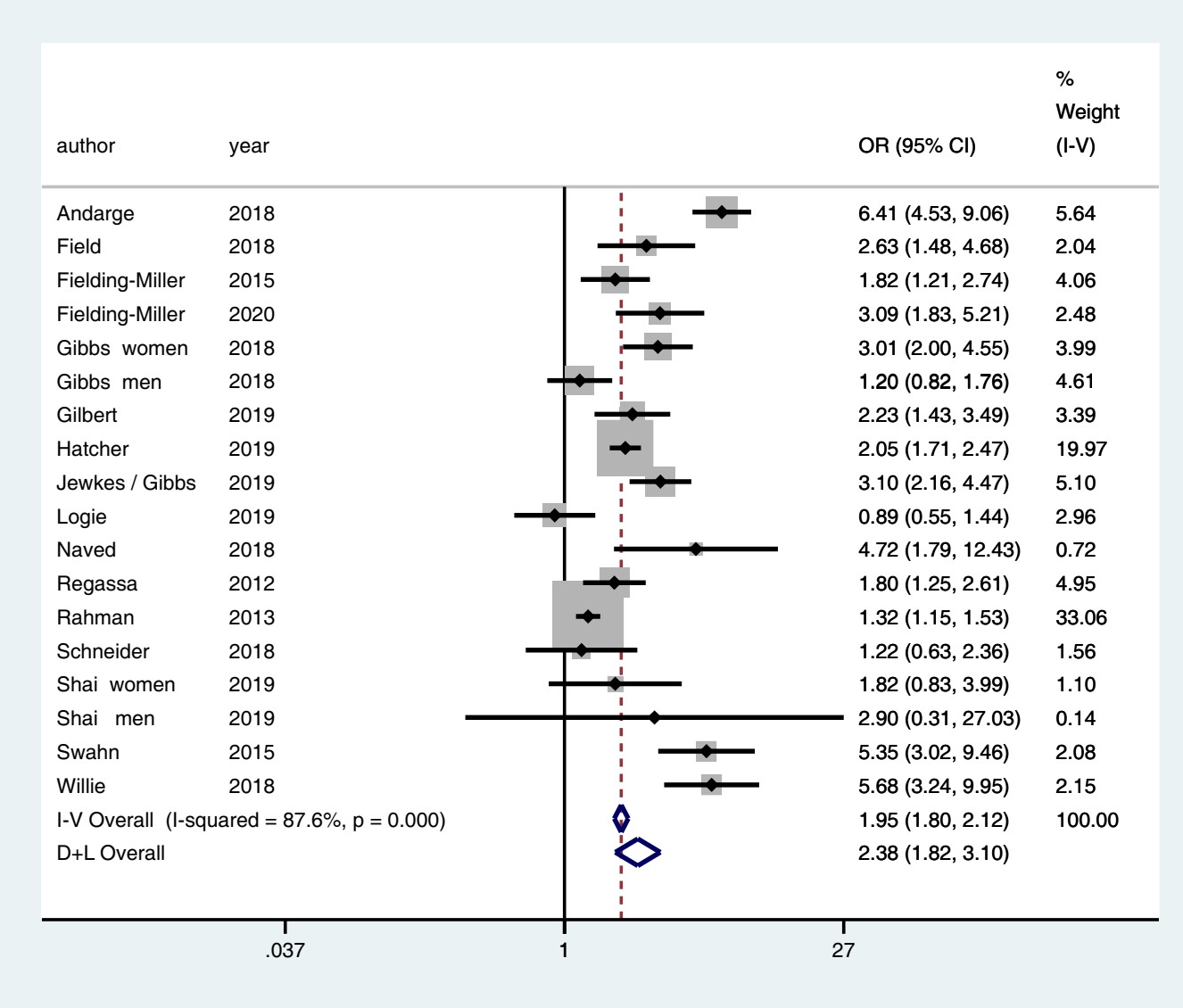

**Fig 2. Meta-analysis of cross-sectional studies of relationship between food security and violence against women and girls.**

VAWG (OR = 1.51, 95%CI = 1.33–1.70). Studies conducted after the *El Nino* year show a stronger association between the two conditions (OR = 2.38, 95%CI = 2.18–2.61).

Meta-analyses were visually inspected for potential publication bias through funnel plots and Egger's test for small-study effects (S2 Fig). There was no evidence of publication bias ($p = 0.09$).

## Qualitative and mixed-methods findings on common drivers and their pathways

The analysis of qualitative and mixed-methods findings suggests that the relationship between food insecurity and VAWG is underpinned by entrenched poverty and gender-inequitable norms. The following describes how the pathways of economic deprivation, alcohol use, and underlying drivers of violence such as gender norms and social ties lead to an increase of

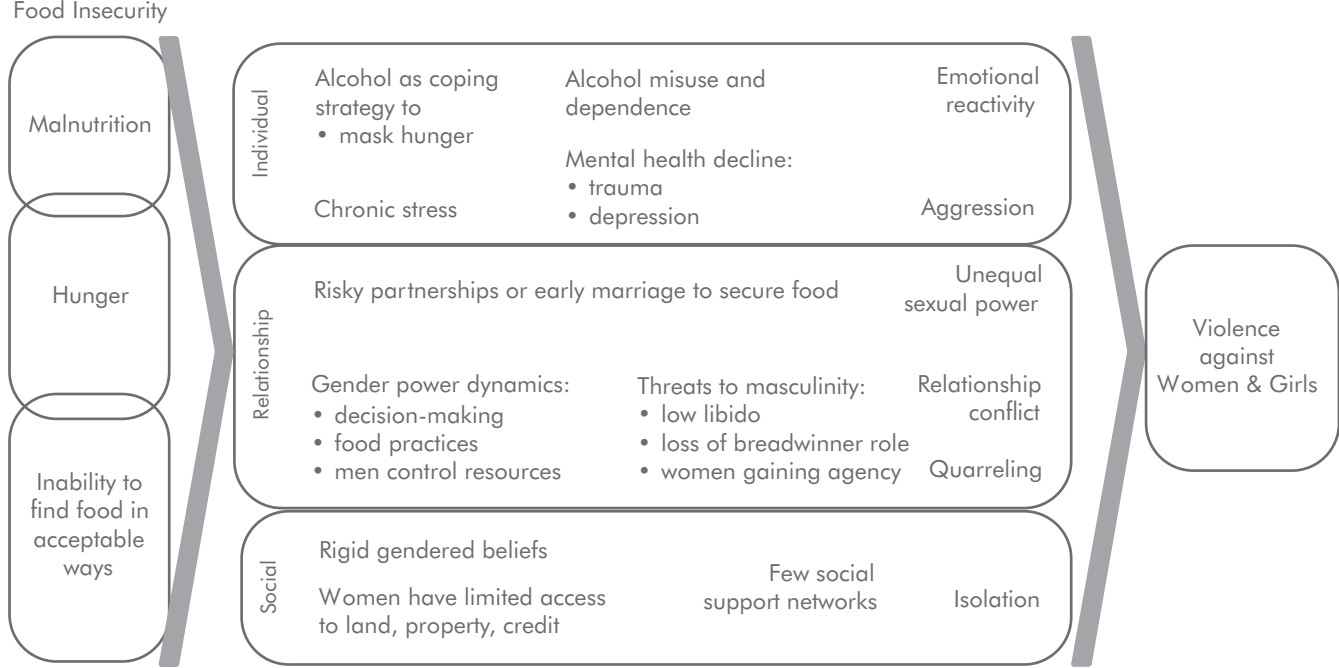

**Fig 3. Conceptual framework of pathways between food insecurity and VAWG.**

VAWG, through routes of individual, relationship, and social behaviours. These potential pathways are highlighted in Fig 3.

**Individual pathways.** **Poor mental health** that results from trauma or chronic stress, especially due to food insecurity, also plays a role in perpetrating VAWG. Among men who are experiencing food insecurity depression seems to be an important pathway leading to VAWG perpetration [28]. Several papers presented the idea that "If the house is not peaceful, food is not eaten" [38]. One mechanism for this may be the psychological distress caused by VAWG exposure. Women reported not eating food because of stress and fear of violence caused by their husband's behaviour [39, 40].

**Alcohol use** may be a maladaptive coping mechanism of being food insecure [12]. Stress and alcohol use depletes psychological resources required to enact self-control over the violent act. Alcohol use may further entrench household conflict if household income is used to buy alcohol instead of purchasing food necessities [41]. Alcohol misuse by men may be related to over-compensating among men who fail to meet the role of provider [42].

**Relationship pathways.** Food insecurity can cause women to engage in **risky survival behaviours**, such as resorting to exploitative transactional sex to meet food needs [38, 41, 43, 44]. In post-conflict settings in Uganda, families sometimes arrange marriages for younger women in their family to secure access to resources, placing the girls at higher risk of violence [38]. Women themselves may strategically chose abusive relationships as an alternative to being without food [39, 41, 43].

**Quarrelling** within the household manifests through conflict over resource allocation [38, 45, 46]. In many settings, men are in a position to control assets and may spend household income on non-food items, including to buy alcohol [45]. Quarrels are also caused by the general stress induced by the lack of food, which compromises the family well-being [46]. If food insecurity is caused by a man losing his job, the financial burden can create relationship conflict and VAWG [41].

Food insecurity can alter **gender power dynamics** in the household. A woman may take on work outside the home to supplement household earnings or men may leave the home to look for work [45]. In either instance of a deepening of traditional roles or a subversion of typical roles, a woman may be at higher risk and further exposed to VAWG [47]. Men may feel their identity as providers are threatened, which can lead to unhealthy coping strategies such as abusing alcohol, taking on multiple partners, or perpetrating VAWG [42, 43]. Women's gendered position as sexually accessible can be compromised if food insecurity leads to a loss of libido [43, 46]. Disempowerment caused by poverty limits the ability of women to circumvent the norms that make them more vulnerable to malnourishment [48].

**Food practices** are also gendered, where being a "woman at the table" means eating later or accessing lower quantity or quality of food. In some settings, women were required to eat after the man and the children, leading some to suggest that women can be exposed to a sort of "food-related violence" [41, 45]. When male partners determine how much money can be allocated to food purchases, their control over women's access to nutrition may similarly be viewed as a form of economic violence [40]. This pattern may explain why studies looking at intra-household malnutrition find that women suffer more than men from under-nutrition [43]. Ager suggested that some women "preferred being beaten [if] children can eat", confirming that strict norms around the ability of women to care for children were sometimes prioritized over individual safety [38, 41].

**Social pathways.**   **Social ties** seem crucial to empower women, but these are challenging to maintain in the context of food insecurity [48–50]. In a study among HIV positive people, lack of food was leading to VAWG partly due to disruptions in social networks due to HIV-related stigma, which distanced people from sources of food support [50].

**Material resources** of land, property and credit can be protective for women. When widows inherit land or other assets, this could lead to increased food security and prevent women from engaging in sexual relationships for food and other goods, as show in [country x] [44]. Women involved in farming cooperatives report improved social assets (networking), human assets (capacity building) and natural assets (access to land) [41]. Other types of empowerment included women's improved decision making [47], managing resources (incl through access to credit) [44] or by attending nutritional sessions or saving groups [13, 42]. In one program in South Africa, there was a new acceptance that there could be a number of income earners in the household and therefore also several decision-makers [48]. Indeed, access to networks is one of the pathways through which successful interventions seem to reduce violence [13, 48, 51].

## Discussion

Evidence from published literature suggests that food insecurity and VAWG are deeply intertwined. In meta-analysis of more than 20,000 participants we found food insecurity is associated with more than double the odds of experiencing or perpetrating VAWG. This finding is consistent across various types of VAWG assessed, study region, and publication timeframe (pre vs. post-*El Nino*). Food insecurity seems to similarly impact women's exposure to VAWG and men's perpetration of this type of violence.

Growing evidence pinpoints why and how food security and VAWG are related. We identified a robust, if relatively small, body of qualitative and mixed methods studies in LMICs. Together, these findings inform a conceptual ecological framework capturing individual, relationship, and social levels. At the individual level, hunger and lack of nutrition can inhibit mental health or lead to maladaptive coping via alcohol use. At the relationship level, economic stress and household conflict over resources can threaten gender roles, aligning with past

(non-systematic) literature reviews [52]. Socially there is an overlap with rigid gender norms that confine women to community positions without access to land, credit/savings, or other material resources. Because food insecurity further isolates women, this has grave implications for their ability to respond to VAWG.

Our evidence synthesis suggests interventions aiming to improve food security and safety for women and girls can operate along three pathways: Addressing mental health and alcohol, shifting power dynamics within the household, and women's social empowerment through material resources and networks. These three layers fit well with promising VAWG prevention intervention research from LMICs. For example, among men, addressing mental health and alcohol misuse seems to be a promising way to limit VAWG perpetration [53–55]. Cash transfers that offer financial stability show promise in reducing couple conflict and household stress [13, 42, 46, 47]. Improving food security may enhance the quality of the couple's relationship in general, and sexual intimacy in particular [46].

An important caveat is the potential for food security interventions to increase VAWG if they lead to backlash by male partners of female participants. If men feel that women have better access to household resources and livelihood opportunities, they may feel their masculinity being threatened and retaliate with violence [47]. Evidence on the frequency of this backlash is not consistent though and recent reviews have not identified it in relation to food insecurity [56–58]. Further, it is plausible that coupling food insecurity provisions with gender empowerment programming may help to reduce or prevent male backlash [47].

## Limitations

The systematic review had several limitations. Papers were only reviewed in English, which limits our ability to draw from papers from all LMIC settings. It is plausible that a larger review inclusive of Spanish and Portuguese language publications would be in a better position to assess the association in that region as only few studies were included that were conducted in Latin America. Because we focused on peer-reviewed literature, we underrepresent perspectives from reports or other grey literature. This may be an acceptable shortcoming for meta-analytic outcomes, but they represent a potentially important gap in our qualitative and mixed methods result. The quantitative studies also used different scales to measure food insecurity and VAWG. It is important to note here that two South American studies included in our review were not included in meta-analysis due to methodologic differences in reporting (and authors were not able to provide additional data by email). We did not abstract information from the studies around research ethics, but given the sensitive nature of the research topics this might be an important area for future reviews to examine more closely.

## Next steps for research, policy, and practice

There is, to date, little evidence of directionality. Since completion of this review, one longitudinal study among men in South Africa suggests food insecurity leads to later VAWG perpetration add reference. The authors found no quantitative association between VAWG perpetration and later food insecurity, though it is plausible that data from women survivors might have distinct findings. The included literature relied on a heterosexual framing of published findings in LMICs. Violence against LGBTIQ communities is higher than in other populations and follows similar patterns of power and control as heterosexual relationships, suggesting food insecurity may be an important dynamic to explore in these samples.

There are major gaps in understanding how policy and broader political forces shape the food security and VAWG intersection. For example, the effect of climate change is likely to be monumental, yet no extant literature, to our knowledge, exists on climate change, food and

VAWG. There is emerging information about Covid-19 and VAWG as well as around Covid-19 and food security. However, we are unaware of publications that combine these three intersecting conditions. There is also a lack of data around how these issues affect young people, a crucial oversight since adolescents and young adults represent a growing proportion of LMIC citizenry.

Evidence for efficacious strategies that jointly target food insecurity and VAWG prevention is limited. This gap is crucial to fill, since these two social drivers of health are highly prevalent in many LMIC communities. Multiple programming approaches have the potential to be sustainable, scalable and cost-effective, but these have limited research demonstrating their efficacy. For example, Gender Action and Learning, a community-based methodology adopted by large-scale organizations such as Oxfam and IFAD, adds gender reflections and visioning exercises to agricultural and financial planning trainings. Gender Champions, a program implemented by Catholic Relief Services, conducts three home visits and implements 'safe spaces' where women and men can discuss common issues. Social Analysis and Action in Food and Nutrition Security, implemented by Care in more than 20 countries, asks community members to take practical steps to address gender, discrimination, or social norms through analysis-action-reflection. According to qualitative or anecdotal evidence, each of these programs seem to improve attitudes about VAWG and decision-making for women, lead to more equitable household division of labour, and offer better economic opportunities. However, none of these has been tested rigorously using experimental study designs, highlighting the importance of funding for research around food and safer relationships programming.

It terms of strategies going forward, programming should be framed as focused on family well-being to benefit households (rather than programming to transform gender). Women's empowerment should be an implicit strategy of the intervention, even if this is not necessarily highlighted during community onboarding. Additionally, improving networks and social ties could be a promising strategy for promoting resilience for both food and violence prevention. Existing innovations in the field should be accompanied by rigorous evidence generation. This can be accomplished by adding VAWG and food security to ongoing trials, carefully evaluating scale-up activities in quasi-experimental design, and funder requirements for practice-based research that includes effectiveness findings.

## Conclusion

This review synthesizes the evidence base on food insecurity and VAWG, and finds these two conditions are statistically and qualitatively related. Both women's experience of VAWG and male perpetration of VAWG are doubled when participants disclose food insecurity. The bidirectionality of these conditions is highlighted in emerging qualitative findings that (1) food insecurity may lead to household stress or tension around gender roles, and/or (2) VAWG exposure may lead to food insecurity if women are more impoverished after leaving a violent household. The review identifies promising strategies for addressing food insecurity and VAWG concurrently: financial stability, gender-transformative approaches, and supporting women's social ties. Global goals to improve nutrition, achieve gender equity, and reduce household poverty will only be accomplished if food security and safer relationships are placed at the forefront of policy and funding decisions in the coming years.

## Supporting information

**S1 Text. Search terms for PubMed, Web of Science, PsycInfo.**
(DOCX)

**S1 Fig. Meta-analysis by sub-groups: Sex, VAWG type, region, study timeframe.**
(DOCX)

**S2 Fig. Funnel plot for publication bias.**
(DOCX)

**S1 Checklist. PRISMA 2009 checklist.**
(PDF)

## Author Contributions

**Conceptualization:** Abigail M. Hatcher, Sabrina Page, Celine Mazars, Heidi Stöckl.

**Data curation:** Abigail M. Hatcher, Lele Aletta van Eck, Rebecca Fielding-Miller.

**Formal analysis:** Abigail M. Hatcher, Sabrina Page, Isabelle Pearson, Celine Mazars.

**Investigation:** Abigail M. Hatcher, Sabrina Page, Celine Mazars, Heidi Stöckl.

**Methodology:** Abigail M. Hatcher, Sabrina Page, Rebecca Fielding-Miller.

**Project administration:** Abigail M. Hatcher, Lele Aletta van Eck, Celine Mazars.

**Resources:** Celine Mazars.

**Software:** Abigail M. Hatcher.

**Supervision:** Heidi Stöckl.

**Validation:** Abigail M. Hatcher, Sabrina Page, Isabelle Pearson.

**Visualization:** Abigail M. Hatcher.

**Writing – original draft:** Abigail M. Hatcher, Celine Mazars.

**Writing – review & editing:** Abigail M. Hatcher, Sabrina Page, Lele Aletta van Eck, Isabelle Pearson, Rebecca Fielding-Miller, Celine Mazars, Heidi Stöckl.

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
